# Effects of Roasting on the Quality of *Moringa oleifera* Leaf Powder and Loaf Volume of *Moringa oleifera*-Supplemented Bread

**DOI:** 10.3390/foods12203760

**Published:** 2023-10-13

**Authors:** Takako Koriyama, Mika Saikawa, Yuria Kurosu, Michiyo Kumagai, Takahiro Hosoya

**Affiliations:** 1Faculty of Food and Nutritional Science, Toyo University, 1-1-1 Izumino, Itakura-machi, Ora-gun, Gunma 374-0193, Japan; mikasaikawa@gmail.com (M.S.); s3c102200029@toyo.jp (Y.K.); hosoya@toyo.jp (T.H.); 2Tokyo Seiei College, 1-4-6 Nishishinkoiwa, Katsushika-ku, Tokyo 124-8530, Japan; kumagai-m@tsc-05.ac.jp

**Keywords:** *Moringa oleifera* leaf powder, *Moringa oleifera*-supplemented bread, loaf volume, roasting, antioxidant activity

## Abstract

Although a decrease in bread volume on adding nutrient-rich *Moringa oleifera* leaf powder (MLP) is known, to our knowledge, improving the swelling of MLP-added bread has not been attempted. This study aimed to investigate the effects of MLP and roasted MLP (RMLP) on bread quality. Bread was supplemented with MLP and RMLP treated at varying temperatures and times; the baked bread was then biochemically evaluated relative to the control. The specific volume of MLP-supplemented bread was 2.4 cm^3^/g, which increased to >4.0 cm^3^/g on using MLP roasted at 130 °C for ≥20 min, demonstrating remarkable swelling. The specific volume of bread supplemented with MLP roasted at 170 °C for 20 min was 4.6 cm^3^/g, similar to that of the control. Additionally, MLP interfered with carbon dioxide production in bread, thus decreasing the abundance of yeast cells; however, RMLP had no such effect and allowed normal fermentation. Scanning electron microscopy revealed gluten formation independent of MLP roasting. Thus, MLP-containing breads generally exhibit suppressed fermentation and expansion due to the bactericidal properties of raw MLP, but these effects are alleviated by heat treatment. These findings highlight the importance of heat treatment in mitigating the effects of MLP on bread fermentation and swelling.

## 1. Introduction

Moringa (*Moringa oleifera*), a plant belonging to the Moringaceae family native to northwestern India, is widely distributed in Africa, South America, and Asia [1,2]. *M. oleifera* is also called the ‘tree of life’ [3,4], as its leaves, seeds, flowers, roots, stems, and sap are utilized for their various health-promoting effects [5,6,7]. In particular, the leaves are rich in protein and minerals, with nine essential amino acids [5,6,8,9], and contain appreciable amounts of phenols and flavonoids, which are used for their antioxidant properties [10,11]. When used as part of a balanced diet, these nutrients scavenge free radicals and may exert immunosuppressive effects [12]. While whole leaves have limited applicability as food ingredients, *M. oleifera* leaf powder (MLP), i.e., ground moringa leaves, is relatively feasible for addition to food. MLP is also rich in proteins and mineral dietary fibers, which is often lacking, and also contains ingredients exhibiting antioxidant activity [11,12,13]. Consequently, the fortification of nutritionally limited staple foods with MLP can compensate for vitamin, protein, and mineral deficiencies [13,14].

Wheat flour bread, a staple that is consumed worldwide as part of the daily diet, is rich in carbohydrates but is low in protein and functional ingredients. As bread is consumed by people of all age groups, increasing its nutrient content would be beneficial for the population. Given the improved living standards and increased public health awareness, various studies have investigated the nutritional enhancement of bread using various plant materials, such as green tea powder [15], celery powder [16], kenaf leaf powder [17], and vegetable leaf powder [18]. The addition of MLP as a dietary supplement to wheat bread and its nutritional benefits have been explored, and most studies recommend adding MLP in the range of 2–2.5% (*w*/*w*) [14,19], mainly because higher MLP content adversely affects bread swelling and palatability. Govender and Siwela [13] reported that bread containing 5% MLP could contribute to addressing malnutrition associated with protein deficiency; unfortunately, the loaf volume of such bread was reduced compared with that of unsupplemented bread. Thus, enhancing the nutritional content of MLP-supplemented bread while preserving loaf volume is a desirable goal. However, to our knowledge, no previous studies have attempted to improve the swelling of bread with added MLP.

Loaf volume is the most important property of bread and provides a quantitative measure of bread baking quality. During the process of bread swelling, carbon dioxide is typically trapped in small air pockets in the dough. During baking, the trapped carbon dioxide causes the air pockets to expand and the starch to gelatinize, transforming the dough into an elastic bread crumb. The dough should remain puffed up when baked, and starch is part of its skeleton [20]. Sengev et al. [4] proposed that the reduction in height and volume of MLP-supplemented bread is caused by an associated reduction in gluten content. Sengev et al. [14] stipulated that the reduction in bread height and volume was the result of the antimicrobial effect of *M. oleifera* leaves on the leavening activity of yeast during dough fermentation. However, the exact reason for the decrease in bread volume upon MLP addition remains unknown.

We hypothesized that if the reduction in bread volume was caused by the chemical components of MLP, the volume and height of bread baked after the removal of these components would remain the same as those of the unsupplemented control. To test this, we inactivated MLP using dry heat treatment and investigated its effect on bread properties, antioxidant capacity, and volume. Under various conditions, dry-roasted MLP samples were added to bread, and their effect on swelling during baking and palatability was observed. Finally, the role of gluten formation in the suppression of swelling owing to the addition of MLP was evaluated.

## 2. Materials and Methods

### 2.1. Roasted Sample Preparation

MLP was purchased from SunRise Ltd. (Okinawa, Japan). The tested roasting temperatures were 110, 130, 150, 170, and 190 °C, and the tested heating times were 10, 20, and 30 min, all processed in an electric oven (NE-BS604, Panasonic, Osaka, Japan). MLP roasting at 170 °C for 30 min and 190 °C for 10 min resulted in burned samples with no material for analysis. Herein, MLP subjected to roasting treatment is referred to as “roasted MLP” (RMLP). The samples were stored in airtight containers at 4 °C and shielded from light until further experiments.

### 2.2. Bread Baking

For MLP-supplemented bread, 5% of the flour weight was substituted with RMLP. The bread was molded and baked using a standard program of the automatic bakery machine (SD-BMT1001; Panasonic, Japan). Briefly, wheat flour (237.5 g), MLP or RMLP (12.5 g), deionized water (180 mL), butter (10 g), salt (5 g), sugar (17 g), non-fat dry milk (6 g), and dry yeast (2.8 g) were added and mixed in the bread maker. MLP was added in powder form to the bread dough just before kneading. The control bread without MLP and RMLP was also prepared at the same time. All ingredients were purchased from a local retail market. The production process required approximately 4 h and included kneading, fermentation, and baking. After baking, the bread was left standing at 25 °C for 90 min, and its physical properties were measured. Samples for analysis were wrapped in a plastic film, placed in a polyethylene bag, and stored at −30 °C until use.

### 2.3. Proximate Analysis

The general nutritional contents of the wheat flour and MLP were analyzed. Moisture, crude protein, crude fat, ash, and crude fiber contents were determined according to the standard methods described by the AOAC [21]. Moisture content was measured using the atmospheric pressure drying method (135 °C). Crude protein content was determined using the Kjeldahl method; the nitrogen-to-protein conversion factor was 6.25. Crude fat content was determined using chloroform–methanol extraction. Dietary fiber contents were determined using the Prosky method. Carbohydrate content was calculated using Equation (1) [22]:(1)Total carbohydrate (g/100 g)=100−[moisture (%)+crude protein (%)+crude fat (%)+ash (%)]

Energy values were calculated using the Atwater system. Mineral components were prepared by the dry-ash method. Sodium, potassium, calcium, magnesium, and iron levels were determined using an atomic absorption spectrophotometer (AA-700; Shimadzu Co., Ltd., Kyoto, Japan). The sample (2.0 g) was mixed with 50 mL of 1.0% hydrochloric acid and stirred for 1 h at 20 °C. The suspension was centrifuged at 3000 rpm for 15 min. Before measurement, the supernatant was filtered through a 0.45 μm nylon filter. Strontium chloride (0.1%) was added to the samples as an interference suppression agent for calcium and magnesium ions.

### 2.4. Water Absorption Capacity

The water absorption capacities of MLP and RMLP samples were measured using the solvent retention capacity profiles [23]. The sample (5.0 ± 0.05 g) was weighed and transferred to a 50 mL centrifuge tube. Distilled water (25 mL) was added to the centrifuge tube and quickly mixed with a vortex mixer for 5 s to homogenize the entire powder with water. The powder was then allowed to solvate and swell for 20 min with shaking for 5, 10, 15, and 20 min. Following this, centrifugation was performed at 1000× *g* for 15 min (Hitachi Koki CR22GIII., Tokyo, Japan). Subsequently, the supernatant was discarded, and the centrifuge tube was placed inverted on a paper towel and allowed to stand for at least 10 min to remove water before weighing. The weight of the plastic 50 mL centrifuge tube was subtracted from the weight of the empty centrifuge tube, and the resulting value was used as the gel weight. The water retention capacity (%) was calculated using Equation (2). The experiment was performed in triplicate.
(2)Water-absorption capacity (%)                  ={(gel wt/flour wt)−1×{86/(100−% flour moisture)}×100

### 2.5. Color Measurements

Color intensity was measured using a spectrophotometer (CM-700d, KONICA MINOLTA, Osaka, Japan) using *L**, *a**, and *b** values according to the CIE color scale. *L** represents the brightness from 0 (black) to 100 (white). The other two coordinates represent redness (+*a**) to greenness (−*a**) and yellowness (+*b**) to blueness (−*b**), respectively. The color difference (Δ*E*) caused by roasting was calculated using Equation (3):(3)$$\Delta E={\left[{\left(L^{*}RMLP-L^{*}MLP\right)}^{2}+{\left(a^{*}RMLP-a^{*}MLP\right)}^{2}+{\left(b^{*}RMLP-b^{*}MLP\right)}^{2}\right]}^{1/2}$$

All experiments were performed in triplicate.

### 2.6. Determination of Physical Properties of Bread

After cooling to 25 °C for 90 min, the bread weight, loaf volume, and crumb texture were measured. Loaf volume was measured using a 3D scanner (Ein Scan SP, Japan 3D Printer, Co., Ltd. Tokyo, Japan), with the turntable step set to 8 and the mesh level set to high. The specific volume of bread was calculated using Equation (4):(4)SV=V/m
where *SV* is the specific volume of the bread (cm^3^/g); *V* is the volume of the bread (cm^3^); and *m* is the mass of the bread (g). All measurements were performed at least in triplicate.

Textural profile analysis was performed using a texture analyzer (TA XT Plus, Stable Micro Systems, Ltd., Godalming, UK). The bread crumb was cut into 2 cm × 2 cm × 2 cm cubes. Each sample was compressed twice until a 70% strain was achieved at a probe speed of 1 mm/s. A cylindrical probe with a 20 mm diameter was used. Crumb hardness (peak force 1), elasticity (length of the base of area 2/length of the base of area 1), and chewiness (hardness × cohesiveness × springiness) were determined as the characteristic parameters of the crumb texture. Six replicates were analyzed for each loaf sample.

### 2.7. Extraction of MLP

Briefly, 1 mL of 80% ethanol was added to 10 mg of MLP or RMLP. The suspension was thoroughly mixed and sonicated for 5 min. After centrifugation (12,054× *g*, 5 min) (CT15RE, HIMAC, Hitachi, Japan), the supernatant was used for further analysis.

### 2.8. DPPH Radical Scavenging Activity

The antioxidant activity was evaluated via a DPPH radical scavenging assay. For the assay, 10 mg/mL of MLP extract was stepwise diluted in 80% ethanol. Next, 100 µL of the stepwise-diluted solution was transferred to a 96-well microplate, and 100 µL of 0.4 mM DPPH radical solution was added. The reaction was allowed to proceed for 30 min at room temperature. Sample absorbance at 520 nm was measured using a microplate reader (Synegy-HTX, Bio Tek, Shoreline, WA, USA) to determine the percentage of DPPH radicals remaining, from which the half-maximal inhibitory concentration (IC_50_, mg/mL) value was calculated.

### 2.9. Determination of Total Polyphenol Content

Total polyphenol content was evaluated using the Folin–Ciocalteu method. Gallic acid used as a standard was dissolved in 80% ethanol, and a series of stepwise dilutions from 2 to 1000 g/mL were prepared. Then, 60 μL of water, 10 μL of stepwise-diluted gallic acid sample, and 15 μL of Folin–Ciocalteu phenol reagent were added to a 96-well plate and incubated for 5 min at room temperature. Next, 75 µL of 2% sodium carbonate was added. After 30 min at room temperature in the dark, sample absorbance at 750 nm was measured using a microplate reader (Synegy-HTX, Bio Tek, USA). A calibration curve was then constructed based on the relationship between absorbance and concentration.

The total polyphenol content of MLP extract was determined in the same manner, with 10 µL of 10 mg/mL MLP extract assayed. The total polyphenol content of MLP extract was calculated as gallic acid equivalents (GAE, mg/mL) based on sample absorbance and the calibration curve for gallic acid.

### 2.10. Quantitative Analysis of the Antioxidant Quercetin-3-Glucoside (Q3G)

High-resolution (HR) electrospray ionization (ESI)–mass spectroscopy (MS) was performed using an Acquity system (Waters, Milford, MA, USA) equipped with a mass spectrometer (Synapt-G2, Waters, USA). Analysis of Q3G was performed using linear gradient elution with solution A (0.1% formic acid in water) and solution B (CH_3_CN containing 0.1% formic acid), delivered at a flow rate of 0.4 mL/min, with a linear gradient from 2% to 100% B over 10 min. The injection volume was 5 μL of sample solution. Data were acquired in the negative MS scanning mode. The mass tolerance of Q3G was within 10 ppm. The Q3G standard was stepwise diluted in ethanol, and the Q3G area under the curve at *m*/*z* 463.0877 and Rt 4.62 min was analyzed by LC/MS. A calibration curve was generated from the relationship between the area value and Q3G concentration. The MLP extract was analyzed by LC/MS under the same conditions, the area value of Q3G was calculated, and the final quantification was performed using the calibration curve.

### 2.11. Fermentation Tests

The fermentation of bread dough was evaluated by measuring the dough volume expansion ratio. The expansion of the dough is attributed to the production of carbon dioxide within the dough during fermentation [24]. The dough was kneaded in an automatic home bakery (SD-BMT1001; Panasonic, Japan) for 20 min using half the amount of ingredients used for bread baking (2.2). The kneaded dough was placed in a bowl and wrapped lightly using plastic wrap, then fermented in an incubator (convection oven, MOV-212F, SANYO, Osaka, Japan) maintained at 38 °C. The dough volume was measured using a 3D scanner (Ein Scan SP; Japan 3D Printer Co., Ltd., Tokyo, Japan) 0, 10, 30, 40, 50, 60, 90, and 120 min after the start of fermentation. The volume expansion ratio was calculated using Equation (5):(5)$$Dough\;volume\;expansion\;ratio=(V_{T}-V_{0})/V_{0}$$
where *V*_0_ refers to the volume at the beginning of fermentation and *V*_T_ indicates the volume after fermentation at T min. All measurements were performed at least in triplicate.

An alcoholic fermentation test was conducted using dry yeast to determine the amount of carbon dioxide produced. The yeast solution was prepared by adding 3 g of dry yeast (Nisshin Super Camellia dry yeast, Nisshin Flour Milling Inc., Tokyo, Japan) to 0.3 g of MLP or 0.3 g of RMLP, mixing well, refrigerating at 4 °C for 2 h, and mixing with 100 mL of distilled water. For the assay, 20 mL of yeast solution and 20 mL of 10% sucrose solution were placed in a Kühne fermentation tube, and the tube opening was covered with non-fat cotton. The tube was placed in a thermostatic chamber (SDN-B, TAITEC, Saitama, Japan) set at 40 °C, and the amount of carbon dioxide produced was read after 20 min. A dry yeast solution prepared without the addition of MLP was used as the control.

### 2.12. Determination of Yeast Viability

Yeast viability was determined using a Cell Counting Kit-8 (WST-8) (Dojindo Laboratories, Kumamoto, Japan). For the assay, 0.30 g of dry yeast, 0.030 g of MLP in 0.30 g of dry yeast, and 0.030 g of RMLP in 0.30 g of dry yeast were mixed and refrigerated at 4 °C for 2 h. Then, 10 mL of purified water was added to each sample and mixed thoroughly. After incubating at room temperature for 30 min, 20 µL of WST-8 reagent was added to 500 µL of each sample, and the mixtures were incubated for 1.5 h at 40 °C. Each solution was centrifuged (12,054× *g* for 5 min), and the absorbance of the supernatant was measured (Synegy-HTX, Bio Tek, USA) at 450 nm.

### 2.13. Scanning Electron Microscopy

Gluten networks in the dough were observed using a scanning electron microscope (JSM-6610LA, JEOL Ltd., Akishima, Japan). The samples were freeze-dried and coated with gold using an ion spatter (E-1010; Hitachi, Tokyo, Japan) for preprocessing. An accelerating voltage of 5 kV was applied.

### 2.14. Sensory Evaluation

The bread crumb was divided into 2 × 2 × 2 cm cubes. A pilot study was conducted before the main study, involving nine participants. Based on the results of the pilot study, the tested roasting conditions were reduced to five for the sensory test, considering the differences between the samples and the ease of comparison: untreated MLP and RMLP obtained by roasting for 20 min at 110 °C, 130 °C, 150 °C, or 170 °C. The sensory evaluation panel consisted of 33 untrained students and faculty (20–65 years old), consisting of 9 males and 24 females, all of whom consumed bread at least once per week. The taste, aroma, appearance, texture, and overall acceptability of the bread were evaluated according to a nine-point hedonic scale (1, extremely dislike; 5, neither like nor dislike; and 9, extremely like). This study was approved by the Toyo University Ethics Committee (approval number TU2022-010).

### 2.15. Statistical Analysis

Data are presented as the mean ± SD. Significant differences in values were analyzed using one-way analysis of variance (ANOVA) in the SPSS statistical software (ver. 27.0, IBM, Armonk, NY, USA). Tukey’s and Bonferroni’s methods were used for multiple comparisons, with statistical significance set at *p* < 0.05. All measurements were performed at least in triplicate.

## 3. Results and Discussion

### 3.1. General Nutritional Composition of MLP

Table 1 shows the results of the proximate analysis of the wheat flour and MLP. The moisture contents of the flour and MLP were 14.3% and 3.7%, respectively, which were significantly different. Hence, the general nutrient ingredient content was expressed as % dry weight. The protein and lipid contents of MLP were higher than those of wheat flour, and MLP contained a large amount of various minerals. Moreover, the dietary fiber content of MLP was 10 times that of wheat flour. MLP was compared with the foods listed in The Standard Tables of Food Composition, Japan (2015) (7th Edition) [25] for specific nutrient contents. The potassium content of MLP was 6.9 times that of bananas; calcium content 19.1 times that of milk; and iron content 4.4 times that of spinach. This is the first time that the nutritional value of MLP from *M. oleifera* grown in Japan has been reported. The nutritional value of Japanese *M. oleifera* leaves was superior to that of other nutritious vegetables and fruits.

### 3.2. Physicochemical Characteristics of RMLP Obtained under Various Roasting Conditions

As shown in Table 2, the moisture content of unroasted MLP was approximately one-third that of wheat flour. The moisture content decreased as the MLP roasting temperature increased and became one-tenth that of wheat flour after roasting at 170 °C for 20 min. The water absorption capacity of unroasted MLP was very high (515%), approximately 4.3 times that of wheat flour (119%). Although this parameter decreased with roasting, it was more than twice that of wheat flour, even for RMLP obtained by roasting at 170 °C for 20 min (244%). The relatively high water absorption of MLP can be explained by its protein, ash, and dietary fiber contents, which were higher than those of wheat flour. The decrease in water absorption after roasting was presumably due to changes in the MLP components and interactions between the components upon heating.

Raw MLP was dark green in color; thus, it had a lower *L** value and more negative *a** value than wheat flour. This was expected because of the high chlorophyll content and natural green color of *M. oleifera* leaves [19,26]. The *L** values of RMLP were lower than those of MLP, and almost all *a** values of RMLP were higher than those of MLP; the difference was more pronounced with an increasing roasting time and temperature. Because *L** indicates lightness and *a** indicates redness, roasting resulted in a darker and redder (i.e., browning) color of MLP. Color degradation occurs when chlorophyll pigments are rapidly converted into pheophytin (a green-grey pigment) during the roasting process. These phenomena were particularly noticeable when roasting at 150 °C for 20 min or longer. In addition, the decrease in *L** value with increasing *a** and *b** values during roasting is reportedly caused by the formation of melanoidin, a brown pigment, a product of the Maillard reaction [27]. Therefore, the color change of MLP upon roasting can be attributed to the non-enzymatic browning associated with Maillard reaction.

### 3.3. Antioxidant Activity of RMLP Obtained under Various Roasting Conditions

Neither DPPH radical scavenging activity nor polyphenols were detected in the control sample (pure wheat flour) (Table 2). In unroasted MLP, the DPPH radical scavenging activity was 1.7 IC_50_ mg/mL, and the phenol content was 149 µg GAE/mL, indicating high antioxidant activity. For RMLP obtained at various roasting temperatures, the IC_50_ values ranged from 1.3 to 2.3 mg/mL, and the total phenol content ranged from 149 to 204 μg GAE/mL. The DPPH radical scavenging activity and polyphenol contents of all RMLPs, except for those obtained by roasting at 170 °C for 20 min, were comparable to or higher than those of unroasted MLP.

We also analyzed the Q3G content of MLP (Table 2). Q3G, a major antioxidant found in Moringa leaves [28], has been validated as a significant contributor to the DPPH radical scavenging activity within MLP through activity-guided fractionation. The Q3G content of MLP prepared under various conditions was analyzed using LC/MS. It was comparable to that of bread prepared using MLP, except for the bread prepared using RMLP by roasting at 170 °C for 20 min, likely due to the Q3G decomposition under these conditions.

*M. oleifera* leaves, both fresh and dried, are an excellent source of antioxidants [19]; however, few studies have investigated the antioxidant activity of roasted MLP. Melanoidin, a brown pigment produced in the late stages of the Maillard reaction, is highly reductive and exhibits strong antioxidant properties [29]. According to previous studies, a positive correlation between color changes related to non-enzymatic browning and total phenol content is observed in many foods [27,30,31]. In this study, a significantly strong positive correlation (*r* = 0.946) was observed between DPPH radical scavenging activity and total phenol content for RMLP obtained by roasting for up to 10 min at 150 °C. These results suggest that the phenolic content increased with the browning of MLP due to roasting. Furthermore, we propose that roasting at 150° C for at least 20 min leads to a simultaneous increase in the number and decomposition of antioxidant substances, and their canceling effect is reflected in the apparent antioxidant activity.

### 3.4. Effect of MLP Roasting Conditions on Bread Loaf Volume

The results of the investigation of the effects of MLP roasting on bread characteristics are shown in Figure 1 and Table 2. The loaf volume and specific volume of bread supplemented with MLP were approximately half those of the no-MLP control, with poor crust formation and caving-like indentations observed at the loaf top. In addition, the hardness of the MLP-supplemented bread was significantly higher than that of the control bread, and the MLP-supplemented bread was less elastic than the control. These observations were consistent with previous findings for wheat bread enriched with 5% MLP [14,19]. In contrast, for bread supplemented with RMLP, the higher the roasting temperature or the longer the roasting time, the significantly higher the specific volume of bread and significantly lower the hardness. A particularly remarkable improvement in swelling was observed with RMLP obtained by roasting at 170 °C for 20 min, with the specific volume of the supplemented bread equal to that of the control and the hardness lower than that of the control. These observations suggested that MLP roasting effectively improves the loaf volume of RMLP-supplemented bread and that higher roasting temperatures improve the swelling of the RMLP-supplemented bread more effectively.

As shown in Table 2, the moisture content of bread was approximately 36.2–38.5%, regardless of the loaf volume, which agrees with the findings of other studies [4,19,32]. This indicates that sufficient moisture evaporation occurs during baking, regardless of the loaf volume. The lightness (*L**) of the crumb and crust decreased significantly with increasing MLP roasting temperature or time. The redness (*a**) of the crumb increased with MLP roasting temperature. The bread color exhibited the same tendency as the powder. Breads with added RMLP obtained by roasting at high temperatures appeared brown.

In general, the volume of bread depends on the distribution and size of air bubbles, which in turn depend on the extensibility of the thin film of gluten. Hence, gluten formation is essential for well-swollen bread. Gluten develops as wheat proteins absorb water, undergo hydration, and interact with one another. Accordingly, one of the reasons for the suppression of swelling in MLP-supplemented bread is the high water absorbency of MLP: wheat proteins are deprived of water that they would normally absorb to form dough gluten, resulting in poorly swollen bread. Nonetheless, although RMLP achieved through roasting at 170 °C for 20 min exhibited lower water absorption than MLP, it still surpassed that of the no-MLP control by more than twofold (Table 2). Hence, additional factors may contribute to the suppression of swelling.

### 3.5. Effect of RMLP on Bread Dough Fermentation

Various fermentation tests were conducted using RMLP obtained by roasting at 170 °C for 20 min, i.e., the roasting condition with the best bread swelling relative to MLP. The effects of roasting on dough volume expansion curves are shown in Figure 2. The control and RMLP-supplemented doughs swelled immediately after the start of fermentation, and the expansion ratios increased with fermentation time. The dough volume expansion ratios of the control and RMLP-supplemented doughs after 120 min of fermentation were similar. In contrast, the MLP (unroasted)-supplemented dough tended to swell significantly less than the other doughs, and its volume was approximately half that of RMLP-supplemented dough after 120 min of fermentation. This trend was consistent with the difference in the specific volumes of *Moringa*-supplemented bread after baking.

Generally, bread swells the most during fermentation and baking [24]. In particular, during fermentation, yeast produces carbon dioxide gas, which creates bubbles and expands the dough. The gluten film spreads around the bubbles formed by the carbon dioxide gas and traps the gas inside the dough so that it does not escape. As the amount of carbon dioxide gas produced increases, the gluten film is pushed outwards from the inside and becomes more flexible [33].

However, in this study, MLP-supplemented dough did not notably swell at the time of fermentation. Therefore, we investigated if poor swelling during fermentation was caused by poor dough formation or poor carbon dioxide production. We investigated alcoholic fermentation and viable yeast counts during dry yeast fermentation. As shown in Figure 3, the amount of carbon dioxide produced by yeast mixed with unroasted MLP was significantly lower than that in the no-MLP control (approximately half that of the control). However, the amount of carbon dioxide produced by yeast mixed with RMLP was higher than that by yeast mixed with MLP and was not significantly different from that in the control. As shown in Figure 4, viable yeast counts were also significantly lower in the presence of unroasted MLP than in the control, whereas the presence of RMLP did not significantly affect the viable yeast counts. The trends in carbon dioxide production and viable yeast counts are consistent.

These findings indicate that MLP kills yeast and inhibits alcoholic fermentation, but these effects are reversed by MLP roasting. This is in agreement with the strong antimicrobial properties of MLP. Aida et al. [34] described that the addition of dry MLP to labneh suppressed the growth of yeast and mold. Similarly, Abd El-Fat et al. [35] reported a lack of yeast or mold growth in cream cheese containing an MLP extract after four weeks of storage. Collectively, the reduction in the height and volume of bread supplemented with MLP is caused by the antimicrobial effect of *M. oleifera* leaves on yeast during fermentation, thereby causing the yeast to become inactive and disabling the complete formation of the gas cells. Of note, although untreated MLP inhibited yeast fermentation, the inhibition was easily ameliorated by MLP heat treatment.

### 3.6. Effect of RMLP on Gluten Formation and Bread Structure

The dough contains a continuous system of gluten proteins with embedded starch, with the gluten proteins creating a fibrous web-like structure (Figure 5A,C). Scanning electron microscopy (SEM) analysis revealed mesh-like gluten formed by globular and fibrous proteins in the control, MLP-supplemented, and RMLP-supplemented dough. However, variations in the network of starch granules and the protein matrix were observed. In particular, the control and RMLP-supplemented dough contained small and large voids with traces of air bubbles in the gluten, whereas the MLP-supplemented dough had a generally compact gluten network structure, with no large voids. No major differences were observed in air bubbles in the different types of bread after baking. These observations indicate that gluten formed normally regardless of MLP supplementation, although the bread containing MLP did not fully expand during baking.

### 3.7. Sensory Evaluation

As shown in Table 3, all MLP-supplemented breads, irrespective of whether they were enriched with MLP or RMLP, were rated 5.7 or above, with no items rated as unfavorable. The different bread samples greatly differed in appearance and texture. High desirability ratings for both were noted for RMLP obtained by roasting at 130 °C and 170 °C, respectively. Particularly, the bread texture obtained higher acceptability ratings for bread made using RMLP obtained by roasting at a high temperature. This indicates that the greatest improvement in desirability related to roasting is associated with texture. The taste of bread supplemented with RMLP obtained by roasting at 130 °C and 170 °C rated higher than that of bread supplemented with unroasted MLP. Among all the tested breads, the bread supplemented with RMLP obtained by roasting at 130 °C for 20 min was the most favored in terms of taste, aroma, appearance, and overall quality. In previous studies, a tendency to not like the green color of Moringa and negative evaluations of its aroma were reported [4]. However, it is thought that Japanese people do not have a sense of objection to the green color and unique scent of Moringa and rather tend to prefer it. This could be because of the Japanese food culture of using green powders such as green tea, matcha, and yomogi. Furthermore, green tea is high in catechins and has high antioxidant activity; therefore, it is thought that Japanese people associate green food that smells of leaves with healthy and nutritious food.

Our findings indicate that supplementing bread with RMLP not only increased the volume but also enhanced palatability. In particular, MLP roasting at 130 °C for 20 min, which made the RMLP-supplemented bread the most palatable, improved bread swelling and increased antioxidant capacity while maintaining Moringa’s unique green color. Roasting MLP at 170 °C for 20 min is recommended for consumers who do not like the green color and strong aroma of Moringa.

## 4. Conclusions

The loaf volume and specific volume of bread supplemented with MLP were reduced by approximately half compared to those of the MLP-free control. However, RMLP-supplemented bread roasted at 130 °C for at least 20 min exhibited improved swelling. Notably, a remarkable enhancement in swelling was observed with RMLP obtained by roasting at 170 °C for 20 min, showing a similar volume to that of the control. After supplementation with MLP and roasting at 170 °C for 20 min, carbon dioxide production was hindered, leading to the killing of approximately half of the yeast cells. In contrast, yeast cells supplemented with RMLP were unaffected, fermenting normally and producing carbon dioxide. SEM observation revealed that gluten formation in the dough occurred irrespective of MLP roasting. These findings imply that the bactericidal component in raw MLP restrained yeast fermentation and impeded bread swelling, and roasting deactivated this component, thereby enabling normal fermentation. These findings underscore the importance of heat treatment in mitigating the effects of MLP on bread fermentation and swelling. In the future, it will be necessary to examine in detail the changes in components caused by heating.

## Figures and Tables

**Figure 1 foods-12-03760-f001:**
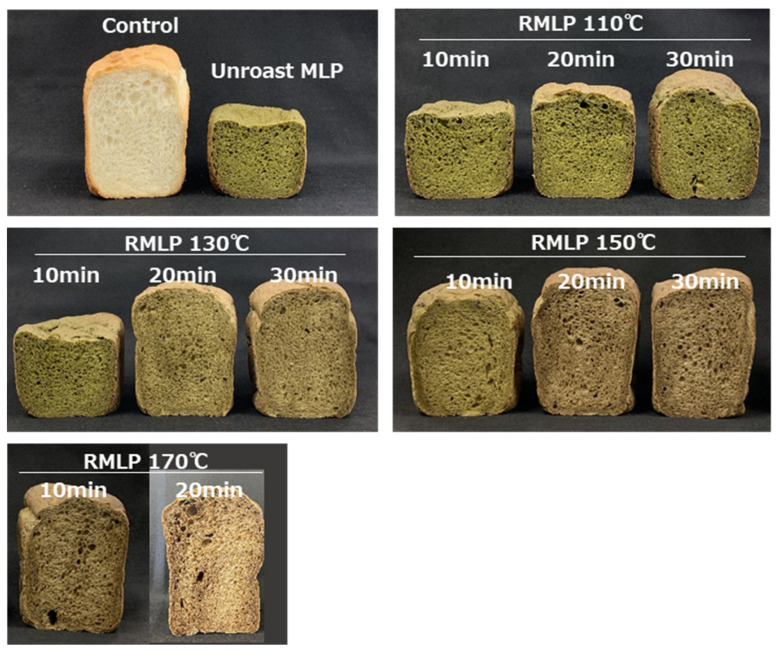
Improvement in bread loaf volume by the addition of RMLP prepared under various roasting conditions. In all samples, MLP or RMLP added to the bread substituted 5% of the wheat flour weight.

**Figure 2 foods-12-03760-f002:**
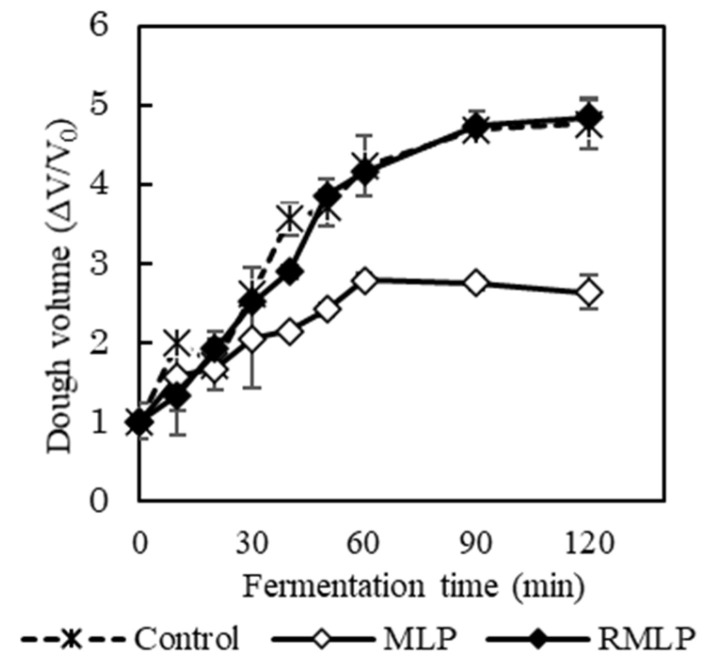
Effect of MLP roasting on the fermentation curve of dough. RMLP was roasted at 170 °C for 20 min. Each bread dough was kneaded in an automatic bakery for 20 min and fermented at 38 °C for 120 min. Dough volume was measured using a 3D scanner. Control, 100% wheat flour bread dough; MLP, bread dough with 5% unroasted MLP; RMLP, bread dough with 5% roasted MLP. Data are expressed as the mean (*n* = 3).

**Figure 3 foods-12-03760-f003:**
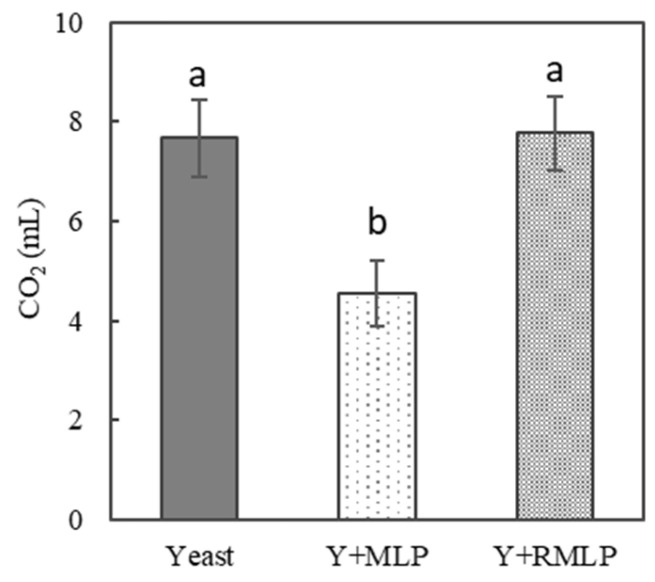
Amount of carbon dioxide generated by alcoholic fermentation using dry yeast. The following yeast solutions were tested: yeast, dry yeast only; Y + MLP, 10% MLP added to dry yeast; Y + RMLP, 10% RMLP added to dry yeast. Significant differences are indicated by different lowercase letters (Tukey’s test, *p* < 0.01; *n* = 3).

**Figure 4 foods-12-03760-f004:**
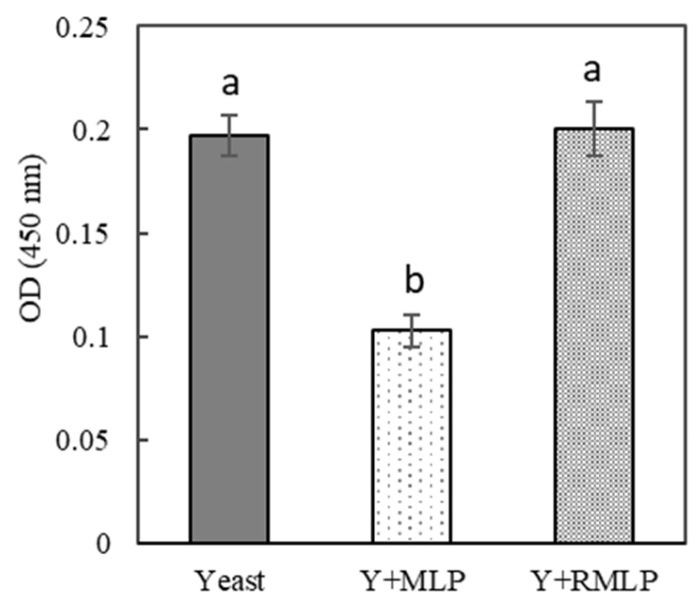
Effect of MLP roasting on dry yeast viability. The following yeast solutions were tested: yeast, dry yeast only; Y + MLP, 10% MLP added to dry yeast; Y + RMLP, 10% RMLP added to dry yeast. Significant differences are indicated by different lowercase letters (Tukey’s test, *p* < 0.01; *n* = 3).

**Figure 5 foods-12-03760-f005:**
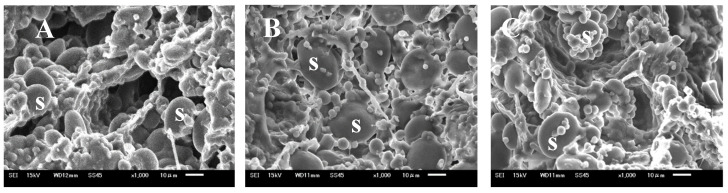
Scanning electron micrographs of different dough and bread samples: (**A**) control dough (no MLP), (**B**) 5% MLP-containing dough, and (**C**) 5% RMLP-containing dough. RMLP was obtained by roasting at 170 °C for 20 min. S: starch.

**Table 1 foods-12-03760-t001:** Nutritional composition of wheat flour and MLP on a dry weight basis (dwb).

Ingredient	Wheat Flour			MLP		
Water content (%)	14.3	±	0.35	3.7	±	0.49
Protein (% dwb)	13.5	±	0.39	27.7	±	8.22
Carbohydrate (% dwb)	84.6	±	1.06	53.2	±	7.95
Dietary fiber (total, % dwb)	3.2	±	0.04	29.1	±	4.34
Dietary fiber (water soluble, % dwb)	1.4	±	0.02	5.1	±	0.75
Dietary fiber (insoluble, % dwb)	1.8	±	0.02	24.0	±	3.59
Crude fat (% dwb)	2.6	±	0.77	8.8	±	1.13
Ash (% dwb)	0.5	±	0.08	10.8	±	0.06
Energy (kcal/100 g dwb)	420	±	9.80	371	±	36.4
Na (mg/100 g dwb)	0	±	0.00	127	±	0.65
K (mg/100 g dwb)	102	±	2.39	2570	±	26.4
Ca (mg/100 g dwb)	20	±	0.46	2024	±	225
Mg (mg/100 g dwb)	26	±	0.62	470	±	74.1
Fe (mg/100 g dwb)	1	±	0.09	10	±	0.54

**Table 2 foods-12-03760-t002:** Effect of roasting treatment on MLP properties and bread quality.

		Wheat Flour	MLP											
			Unroasted	110 °C			130 °C			150 °C			170 °C	
				10 min	20 min	30 min	10 min	20 min	30 min	10 min	20 min	30 min	10 min	20 min
**Flour**														
Moisture content (%) Water absobing capacity (%)		14.5 ± 0.01 ^a^	4.9 ± 0.31 ^b^	3.7 ± 0.11 ^c^	2.7 ± 0.25 ^d^	1.8 ± 0.26 ^e^	2.1 ± 0.23 ^de^	1.8 ± 0.25 ^e^	1.6 ± 0.41 ^e^	1.9 ± 0.30 ^de^	1.9 ± 0.23 ^e^	1.8 ± 0.20 ^e^	2.1 ± 0.46 ^de^	1.5 ± 0.50 ^e^
		119 ± 6.77 ^g^	515 ± 17.2 ^ab^	464 ± 22.1 ^ab^	465 ± 18.6 ^ab^	461 ± 22.7 ^ab^	506 ± 16.1 ^a^	405 ± 20.3 ^bcd^	341 ± 15.1 ^cde^	411 ± 24.2 ^bc^	314 ± 6.65 ^ef^	323 ± 3.84 ^def^	342 ± 22.9 ^cde^	244 ± 13.1 ^f^
Color	L*	90.1 ± 0.01 ^a^	57.9 ± 0.68 ^b^	56.4 ± 0.06 ^c^	54.9 ± 0.05 ^d^	53.9 ± 0.05 ^e^	53.1 ± 0.06 ^e^	48.7 ± 0.03 ^fg^	48.3 ± 0.26 ^g^	49.5 ± 0.05 ^f^	39.4 ± 0.19 ^h^	38.1 ± 0.06 ^i^	40.0 ± 0.06 ^h^	32.8 ± 0.20 ^j^
	a*	−0.8 ± 0.10 ^f^	−12.7 ± 0.13 ^m^	−13.2 ± 0.11 ^l^	−10.6 ± 0.07 ^k^	−7.3 ± 0.06 ^i^	−9.9 ± 0.16 ^j^	−2.1 ± 0.08 ^g^	0.3 ± 0.04 ^e^	−2.4 ± 0.05 ^h^	4.3 ± 0.09 ^c^	4.7 ± 0.01 ^b^	3.0 ± 0.04 ^d^	5.5 ± 0.07 ^a^
	b*	9.72 ± 0.05 ^h^	37.2 ± 1.01 ^b^	38.4 ± 0.08 ^a^	37.1 ± 0.21 ^b^	35.2 ± 0.07 ^c^	36.6 ± 0.14 ^b^	32.3 ± 0.06 ^d^	31.4 ± 0.21 ^d^	34.6 ± 0.03 ^e^	26.9 ± 0.07 ^ef^	25.5 ± 0.03 ^f^	27.1 ± 0.03 ^e^	22.3 ± 0.19 ^g^
	ΔE	—	—	1.8 ± 0.07	3.2 ± 0.03	6.7 ± 0.09	6.4 ± 0.17	15.5 ± 0.06	17.7 ± 0.25	13.9 ± 0.07	27.6 ± 0.21	29.1 ± 0.05	26.3 ± 0.05	28.7 ± 7.91
DPPH radical scavenging activity (IC_50_, mg/mL)		N/A	1.7 ± 0.07 ^b^	1.6 ± 0.05 ^bc^	1.7 ± 0.08 ^b^	1.4 ± 0.07 ^de^	1.6 ± 0.05 ^bcd^	1.4 ± 0.03 ^cde^	1.4 ± 0.06 ^cde^	1.3 ± 0.06 ^e^	1.6 ± 0.04 ^bcde^	1.7 ± 0.05 ^b^	1.4 ± 0.05 ^e^	2.3 ± 0.07 ^a^
Total phenol content (μg GAE/mL)		N/A	149 ± 2.75 ^d^	152 ± 9.96 ^d^	149 ± 7.77 ^d^	173 ± 6.66 ^bcd^	159 ± 6.06 ^cd^	178 ± 5.01 ^bc^	180 ± 21.18 ^abc^	187 ± 5.39 ^ab^	174 ± 0.13 ^bcd^	161 ± 4.92 ^bcd^	204 ± 9.29 ^a^	123 ± 6.71 ^e^
Q3G content (mg/g)		N/A	10.3 ± 1.11 ^abcd^	9.3 ± 0.96 ^cd^	9.7 ± 1.11 ^bcd^	10.6 ± 0.85 ^abcd^	10.4 ± 0.57 ^abcd^	11.5 ± 0.57 ^abc^	12.0 ± 0.58 ^ab^	12.3 ± 0.87 ^a^	10.3 ± 0.83 ^abcd^	8.7 ± 1.11 ^d^	12.4 ± 1.21 ^a^	1.9 ± 0.47 ^e^
**Bread** ^(1)^														
Moisture content (%)		38.5 ± 0.24 ^a^	37.1 ± 0.88 ^abcd^	37.4 ± 0.42 ^abcd^	36.6 ± 1.39 ^cd^	36.2 ± 0.38 ^d^	37.1 ± 1.09 ^abcd^	37.1 ± 1.19 ^abcd^	37.8 ± 0.17 ^abc^	38.2 ± 0.27 ^ab^	37.1 ± 0.50 ^abcd^	37.5 ± 0.21 ^abcd^	36.7 ± 0.93 ^bcd^	37.4 ± 0.30 ^abcd^
Loaf volume (cm^3^)		1844 ± 40.2 ^a^	990 ± 41.3 ^j^	1002 ± 42.1 ^i^	1250 ± 43.2 ^g^	1389 ± 45.4 ^f^	1118 ± 44.9 ^h^	1624 ± 56.6 ^d^	1703 ± 52.1 ^c^	1545 ± 47.8 ^e^	1818 ± 58.7 ^b^	1763 ± 42.6 ^bc^	1723 ± 52.7 ^c^	1856 ± 59.9 ^a^
Specific volume (cm^3^/g)		4.6 ± 0.12 ^a^	2.4 ± 0.1 ^f^	2.5 ± 0.02 ^f^	3.1 ± 0.11 ^e^	3.4 ± 0.03 ^d^	2.8 ± 0.08 ^ef^	4.0 ± 0.02 ^c^	4.2 ± 0.07 ^bc^	3.8 ± 0.12 ^c^	4.4 ± 0.03 ^b^	4.4 ± 0.03 ^b^	4.3 ± 0.05 ^b^	4.6 ± 0.08 ^a^
Hardness (N)		3.5 ± 0.91 ^bef^	26.8 ± 3.47 ^a^	18.2 ± 3.56 ^b^	14.5 ± 5.13 ^c^	7.2 ± 1.9 ^de^	7.5 ± 1.51 ^d^	3.1 ± 0.55 ^def^	3.4 ± 0.68 ^def^	3.6 ± 0.47 ^def^	3.2 ± 0.79 ^def^	2.8 ± 1.04 ^def^	2.5 ± 0.32 ^ef^	2.2 ± 0.22 ^f^
Elasticity		1.2 ± 0.25 ^ab^	0.9 ± 0.02 ^b^	0.9 ± 0.05 ^b^	0.9 ± 0.02 ^ab^	1.0 ± 0.01 ^ab^	1.0 ± 0.01 ^ab^	1.0 ± 0.04 ^a^	1.0 ± 0.01 ^ab^	1.0 ± 0.01 ^ab^	1.1 ± 0.12 ^ab^	1.0 ± 0.02 ^ab^	1.0 ± 0.03 ^ab^	1.1 ± 0.03 ^ab^
Chewiness		3.2 ± 0.88 ^c^	11.9 ± 2.33 ^a^	9.9 ± 2.06 ^a^	7.8 ± 2.63 ^b^	4.6 ± 0.91 ^bc^	4.6 ± 1.09 ^bc^	2.7 ± 1.76 ^c^	2.6 ± 0.52 ^c^	2.1 ± 0.34 ^c^	2.4 ± 0.36 ^c^	2.6 ± 0.62 ^c^	2.3 ± 0.71 ^c^	1.9 ± 0.64 ^c^
VRC ^(2)^%		0.4 ± 0.03 ^a^	0.2 ± 0.01 ^g^	0.3 ± 0.06 ^fg^	0.2 ± 0.04 ^ef^	0.2 ± 0.0 ^cde^	0.3 ± 0.03 ^bc^	0.3 ± 0.02 ^b^	0.3 ± 0.02 ^b^	0.3 ± 0.01 ^ab^	0.3 ± 0.01 ^bcd^	0.3 ± 0.01 ^bcd^	0.3 ± 0.02 ^bc^	0.3 ± 0.01 ^ab^
Crumb color	L*	79.1 ± 0.76 ^a^	42.6 ± 1.39 ^cd^	41.7 ± 0.25 ^de^	45.0 ± 0.64 ^bcd^	46.0 ± 0.77 ^bc^	43.6 ± 1.32 ^cd^	46.0 ± 0.99 ^bc^	47.0 ± 0.67 ^b^	46.1 ± 0.09 ^bc^	42.9 ± 2.18 ^de^	40.9 ± 0.81 ^e^	41.5 ± 0.83 ^de^	40.8 ± 1.39 ^de^
	a*	−0.9 ± 0.27 ^de^	−1.9 ± 0.85 ^e^	−0.9 ± 0.34 ^e^	−1.0 ± 0.39 ^e^	−1.0 ± 0.11 ^e^	−0.3 ± 0.15 ^de^	1.5 ± 0.74 ^c^	1.9 ± 0.11 ^bc^	0.9 ± 0.65 ^cd^	4.5 ± 0.48 ^a^	4.6 ± 0.34 ^ab^	3.1 ± 0.26 ^ab^	5.07 ± 0.65 ^a^
	b*	17.5 ± 0.32 ^f^	32.9 ± 0.72 ^a^	32.3 ± 1.26 ^ab^	32.9 ± 0.33 ^ab^	31.5 ± 0.78 ^ab^	30.4 ± 1.02 ^bc^	25.1 ± 2.49 ^d^	24.5 ± 1.13 ^d^	28.2 ± 1.21 ^c^	20.7 ± 0.99 ^e^	23.2 ± 0.36 ^de^	23.2 ± 0.36 ^de^	15.5 ± 0.46 ^f^
	ΔE	—	—	1.7 ± 0.77	3.5 ± 0.58	4.8 ± 0.94	4.3 ± 1.34	10.1 ± 2.11	11.1 ± 1.18	8.3 ± 0.91	14.4 ± 0.77	12.1 ± 0.43	26.3 ± 0.05	19.3 ± 1.53
Crust color	L*	56.4 ± 2.27 ^a^	35.7 ± 0.33 ^c^	37.1 ± 4.22 ^c^	36.8 ± 3.53 ^c^	36.0 ± 3.07 ^c^	36.0 ± 3.05 ^c^	35.8 ± 1.68 ^c^	40.8 ± 0.98 ^bc^	37.0 ± 3.18 ^bc^	36.8 ± 2.11 ^c^	38.6 ± 1.41 ^bc^	42.3 ± 2.35 ^b^	39.5 ± 1.94 ^bc^
	a*	14.1 ± 1.06 ^a^	7.1 ± 0.58 ^def^	5.1 ± 1.23 ^def^	4.4 ± 1.99 ^f^	4.5 ± 1.19 ^ef^	6.3 ± 0.89 ^cdef^	7.1 ± 0.58 ^bc^	6.6 ± 1.26 ^cdef^	7.0 ± 0.7 ^bcd^	8.7 ± 1.54 ^b^	6.7 ± 1.17 ^bcde^	7.7 ± 0.95 ^bc^	6.9 ± 0.35 ^bcde^
	b*	32.7 ± 1.08 ^a^	21.5 ± 0.67 ^bc^	17.1 ± 2.79 ^d^	18.0 ± 2.84 ^cd^	20.1 ± 1.84 ^bcd^	19.9 ± 0.54 ^bcd^	21.4 ± 2.04 ^bc^	23.9 ± 0.85 ^b^	22.7 ± 2.21 ^b^	20.1 ± 1.86 ^bcd^	20.0 ± 2.41 ^bcd^	23.2 ± 1.62 ^b^	18.2 ± 0.93 ^cd^
	ΔE	—	—	6.5 ± 0.58	6.3 ± 1.574	4.3 ± 1.54	2.5 ± 1.98	2.1 ± 1.65	5.4 ± 1.28	3.0 ± 2.65	4.0 ± 1.02	4.2 ± 1.16	6.8 ± 2.21	5.3 ± 1.33

^(1)^ Bread, bread containing each of the above powders. In all cases, the added MLP or RMLP substituted 5% of the wheat flour weight. ^(2)^ VRC, volume recovery coefficient. Values with different letters in the same row are significantly different (Bonferroni’s test, *p* < 0.05; *n* = 5).

**Table 3 foods-12-03760-t003:** Sensory evaluation of desirability of bread containing RMLP prepared under different roasting conditions.

MLP		Taste			Aroma			Appearance			Texture			Overall		
Unroasted		6.7	±	1.6	6.3	±	1.8	7.1	±	1.4 ^a^	6.5	±	1.8 ^bc^	6.8	±	1.4
Roasted	110 °C	6.3	±	2.0	6.6	±	1.7	6.9	±	1.5 ^ab^	6.3	±	2.1 ^c^	6.3	±	1.9
	130 °C	7.3	±	1.5	6.6	±	1.8	7.2	±	1.5 ^a^	7.5	±	1.3 ^ab^	7.0	±	1.5
	150 °C	6.3	±	1.9	5.7	±	2.0	6.0	±	1.8 ^b^	7.5	±	1.6 ^ab^	6.4	±	2.0
	170 °C	7.1	±	1.6	5.8	±	2.1	6.1	±	2.0 ^ab^	7.9	±	1.0 ^a^	6.8	±	1.7

Values with different letters in the same row are significantly different (Tukey’s test, *p* < 0.05; *n* = 33).

## Data Availability

The data used to support the findings of this study can be made available by the corresponding author upon request.

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
