# Peer review of "Effects of Roasting on the Quality of Moringa oleifera Leaf Powder and Loaf Volume of Moringa oleifera-Supplemented Bread"

_foods, 2023, doi:10.3390/foods12203760_

Round 1

Reviewer 1 Report

I am very grateful you for the invitation to review manuscript foods-2650408 by Koriyama and coauthors "Effects of roasting on the quality of Moringa oleifera leaf powder and loaf volume of Moringa-oleifera-supplemented bread”. The present study aimed to evaluate the impact of Moringa oleifera leaf powder and roasted Moringa oleifera leaf powder on bread quality. The work is interesting but needs adjustments to increase the quality of the material.

Comments:

- Abstract, line 11: What benefits? To specify.

- Lines 12-13: What did the studies evaluating the addition in bread evaluate?

- Abstract: Please indicate a better step-by-step about the work, including what was evaluated and the conditions.

- Abstract: Present the most specific results. Insert numerical results related to the main findings of the work. A better explanation of what "notable", "reduction", and “normal” should be given.

- Line 22: Change the repeated keywords by different words from the title.

- Lines 29-31: Present the average composition of nutritional components.

- Line 34: Specify the health benefits.

- Introduction: The production, market, and consumption aspects of the vegetable must be deepened.

- Introduction: Safety aspects must be discussed, since, in Brazil, for example, the consumption of Moringa is prohibited due to the lack of safety evidence.

- Line 214: Insert the approved protocol number.

- Lines 225-227: Insert numeric results for easy comparison.

- 3.1. General nutritional composition of MLP: Specify the amount needed to reach the RDA.

- Line 246: The water absorption capacity was not presented in M.M. section.

- The material and methods section must be restructured, following the order of execution of the work, as well as the analytical part.

- 3.4. Effect of MLP roasting conditions on bread loaf volume: Deepen the chemical/biochemical and technological issues of dough formation.

Author Response

Dear Reviewer 1:

We are pleased to submit a revised version of our manuscript, “Effects of roasting on the quality of Moringa oleifera leaf powder and loaf volume of Moringa-oleifera-supplemented bread” (foods-2650408).

A point-by-point response has been provided for each query, and the corresponding changes in the main manuscript have been highlighted in yellow.

Comments:

Abstract, line 11: What benefits? To specify.

Lines 12-13: What did the studies evaluating the addition in bread evaluate?

Responses:

Thank you for the opportunity to provide clarity here. For better readability, we have deleted the statements “Consumption of Moringa oleifera leaf powder (MLP) is associated with health benefits. Studies have shown that bread containing MLP has adversely affected its loaf volume. However, studies have not addressed this issue”.

Comments:

Abstract: Please indicate a better step-by-step about the work, including what was evaluated and the conditions.

Abstract: Present the most specific results. Insert numerical results related to the main findings of the work. A better explanation of what "notable", "reduction", and “normal” should be given.

Responses:

 We apologize for the lack of specificity in the abstract. In view of your comment, we have revised the abstract to present information in a more specific and sequential manner. Following is the edited abstract:

“Although a decrease in bread volume on adding nutrient-rich Moringa oleifera leaf powder (MLP) is known, to our knowledge, improving the swelling of MLP-added bread has not been attempted. This study aimed to investigate the effects of MLP and roasted MLP (RMLP) on bread quality. Bread was supplemented with MLP and RMLP treated at varying temperatures and time; the baked breads were then biochemically evaluated relative to the control. The specific volume of MLP-supplemented bread was 2.4 cm3/g, which increased to >4.0 cm3/g on using MLP roasted at 130 ℃ for ≥20 min, demonstrating remarkable swelling. The specific volume of bread supplemented with MLP roasted at 170 °C for 20 min was 4.6 cm3/g, similar to that of the control. Additionally, MLP interfered with carbon dioxide production in bread, thus decreasing the abundance of yeast cells; however, RMLP had no such effect and allowed normal fermentation. Scanning electron microscopy revealed gluten formation independent of MLP roasting. Thus, MLP-containing breads generally exhibit suppressed fermentation and expansion due to the bactericidal properties of raw MLP, but these effects are alleviated by heat treatment. These findings highlight the importance of heat treatment in mitigating the effects of MLP on bread fermentation and swelling.

Comment:

Line 22: Change the repeated keywords by different words from the title.

Response:

We appreciate the careful reading. The revised Keywords have been provided below:

Keywords: Moringa oleifera leaf powder; Moringa oleifera-supplemented bread; loaf volume; roasting; antioxidant activity

Comment:

  Lines 29-31: Present the average composition of nutritional components.

Response:

  Thank you for your suggestion. The average values have been presented in Table 1 

Comment:

Line 34: Specify the health benefits.

Response:

  Thank you for the opportunity to provide clarity here. In view of your comment, we have revised the text as mentioned below.

“While whole leaves have limited applicability as food ingredients, M. oleifera leaf powder (MLP), i.e., ground moringa leaves, is relatively feasible for addition to food. MLP is also rich in proteins and mineral dietary fibers, which is often lacking, and also contains ingredients exhibiting antioxidant activityï¼»11-13ï¼½. Consequently, the fortification of nutritionally limited staple foods with MLP can compensate for vitamin, protein, and mineral deficiencies [13,14].”

Comment:

Introduction: The production, market, and consumption aspects of the vegetable must be deepened.

Response:

  Unfortunately, moringa is still in the process of commercialization in Japan. Hence, limited literature is available on these aspects.

Comment:

Introduction: Safety aspects must be discussed, since, in Brazil, for example, the consumption of Moringa is prohibited due to the lack of safety evidence.

Response:

Thank you for this constructive suggestion. This study was conducted in Japan, where no such restrictions currently exist. However, in view of your comment, we intend to determine the safety of moringa's ingredients and present the data in near future.

Comment:

Line 214: Insert the approved protocol number.

Response:

  Thank you for the careful reading. As suggested, we have added the approval number in the manuscript as follows:

“This study was approved by the Toyo University Ethics Committee (approval number TU2022-010).”

Comment:

Lines 225-227: Insert numeric results for easy comparison.

Response:

  As suggested by you, we have inserted the numeric results in the sentence as mentioned below.

“The moisture contents of the flour and MLP were 14.5% and 4.9%, respectively, which were significantly different. Hence, the general nutrient ingredient content was expressed as % dry weight.”.

Comment:

3.1. General nutritional composition of MLP: Specify the amount needed to reach the RDA.

Response:

  We value your suggestion. However, RDA values are not listed in this manuscript because they differ not only by country but also by age and gender.

Comment:

Line 246: The water absorption capacity was not presented in M.M. section.

Response:

 We apologize for the missing information. Accordingly, we have added details regarding the water-absorption capacity in the Methods section as follows:

“2.4. Water-absorption capacity

The water-absorption capacities of MLP and RMLP samples were measured using the solvent retention capacity profiles (AACC method 56-11). Sample (5.0 ± 0.05 g) was weighed and transferred to a 50 mL centrifuge tube. Distilled water (25 mL) was added to the centrifuge tube and quickly mixed with a vortex mixer for 5 s to homogenize the entire powder with water. The powder was then allowed to solvate and swell for 20 min with shaking for 5, 10, 15, and 20 min. Following this, centrifugation was performed at 1,000 × g for 15 min (Hitachi Koki CR22GIII, Japan). Subsequently, the supernatant was discarded, and the centrifuge tube was placed inverted on a paper towel and allowed to stand for at least 10 min to remove water before weighing. The weight of the plastic 50 mL centrifuge tube was subtracted from the weight of the empty centrifuge tube, and the resulting value was used as the gel weight. The water retention capacity (%) was calculated using Eq. (2). The experiment was performed in triplicate.

Water-absorption capacity (%) = {(gel wt / flour wt)-1 × {86/(100-% flour moisture)} ×100

(2)”

Comment:

The material and methods section must be restructured, following the order of execution of the work, as well as the analytical part.

Response:

 Thank you for the constructive suggestion. In view of your comment, we have restructured the Methods section as well as the analytical section.

Comment:

3.4. Effect of MLP roasting conditions on bread loaf volume: Deepen the chemical/biochemical and technological issues of dough formation.

Response:

 We are currently analyzing the components altered on heating MLP and conducting further studies to explore this information; we intend to publish the findings as a research paper in future.

Thank you again for the thoughtful suggestions and insights, which have enriched the manuscript and produced a better and more balanced account of the research.

Reviewer 2 Report

This is an interesting paper presenting the possibility of including MLP in bread and the use of heat treatment to counteract the negative effects of this ingredient. Some corrections are needed and the suggestions are presented below.  

English needs revision.

Abstract: "Studies have shown that bread containing MLP has adversely affected its loaf volume. However, 12 studies have not addressed this issue." This is contradictory. What do you mean?

The methods are not mentioned in the abstract.

Please add some numerical data to the results in the abstract.

Introduction:

L 50 There is a piece of confusing information. The authors mention before that the supplementation of bread with MLP has been already studied, but here they affirm that there are not studies regarding the effects on the loaf volume. Please clarify; maybe there are not studies at doses >2.5%.

Materials and methods:

Please include a section of Materials and describe the origin of the materials used.

L76 What about 190 degrees? The sample was ok?

L81 Have you ground the MLP? Please explain in which form it has been added to the bread.

L88 Please give details about the fermentation and baking conditions. 

L100/112/178 Please write with the Equation editor.

L103 Please give details about the AAS method.

L110 This is not the hue angle, but the color difference (delta E). Please revise in the whole manuscript and replace hue with color difference.

L136 Please give the dilutions used.

L200 Have you extracted the gluten or have you analysed the dough? Please clarify.

Results and discussion:

Please explain why from section 3.4 you chose only 1 treatment condition for each addition. Furthermore, in the sensory analysis, you included more samples that in the sections 3.4-3.6. Why?

All the results should be compared and explained by using other similar findings from the literature. 

Conclusions:

Please mention the limits and further research perspectives.

English needs some revision. 

Author Response

Dear Reviewer 2:

We are pleased to submit a revised version of our manuscript, “Effects of roasting on the quality of Moringa oleifera leaf powder and loaf volume of Moringa-oleifera-supplemented bread” (foods-2650408).

A point-by-point response has been provided for each query, and the corresponding changes in the main manuscript have been highlighted in yellow.

Comments:

Abstract: "Studies have shown that bread containing MLP has adversely affected its loaf volume. However, 12 studies have not addressed this issue." This is contradictory. What do you mean?

Please add some numerical data to the results in the abstract.

Responses:

 We apologize for the lack of clarity. In view of your comment, we have removed the discrepant information and revised the abstract as follows:

“Although a decrease in bread volume on adding nutrient-rich Moringa oleifera leaf powder (MLP) is known, to our knowledge, improving the swelling of MLP-added bread has not been attempted. This study aimed to investigate the effects of MLP and roasted MLP (RMLP) on bread quality. Bread was supplemented with MLP and RMLP treated at varying temperatures and time; the baked breads were then biochemically evaluated relative to the control. The specific volume of MLP-supplemented bread was 2.4 cm3/g, which increased to >4.0 cm3/g on using MLP roasted at 130 ℃ for ≥20 min, demonstrating remarkable swelling. The specific volume of bread supplemented with MLP roasted at 170 °C for 20 min was 4.6 cm3/g, similar to that of the control. Additionally, MLP interfered with carbon dioxide production in bread, thus decreasing the abundance of yeast cells; however, RMLP had no such effect and allowed normal fermentation. Scanning electron microscopy revealed gluten formation independent of MLP roasting. Thus, MLP-containing breads generally exhibit suppressed fermentation and expansion due to the bactericidal properties of raw MLP, but these effects are alleviated by heat treatment. These findings highlight the importance of heat treatment in mitigating the effects of MLP on bread fermentation and swelling.”

Comment:

L 50 There is a piece of confusing information. The authors mention before that the supplementation of bread with MLP has been already studied, but here they affirm that there are not studies regarding the effects on the loaf volume. Please clarify; maybe there are not studies at doses >2.5%.

Response:

 We apologize for the discrepancy. Accordingly, we have revised the relevant text as follows: “However, to our knowledge, no previous studies have attempted to improve the swelling of bread with added MLP.”.

Comment:

L76 What about 190 degrees? The sample was ok?

Response:

  On treatment at 190 degrees, the sample was burnt and unfit for analysis. The same has already been mentioned in section “2.1. Roasted sample preparation” Lines 82-84 as follows:

MLP roasting at 170 °C for 30 min and 190 °C for 10 min resulted in burned samples with no material for analysis (data not shown).”

Comment:

L81 Have you ground the MLP? Please explain in which form it has been added to the bread.

Response:

 As mentioned in section “2.1. Roasted sample preparation” Line 79, we have purchased M. oleifera leaf powder (MLP) and used it for addition to bread. The same has been mentioned in Section 2.1 as follows:

MLP was purchased from SunRise Ltd. (Okinawa, Japan).”

Additionally, the following text has been added to provide a more detailed explanation Line 92-92, “MLP was added in powder form to the bread dough just before kneading”.

Comment:

L88 Please give details about the fermentation and baking conditions.

Response:

  Thank you for the opportunity to provide clarity here. Initially, the dough was kneaded and was left to rest to homogenize the flour and water. The dough was then kneaded again, fermented, and baked. The entire process lasted for 4 h and was performed automatically using the bakery machine (SD-BMT1001; Panasonic, Japan).

Comment:

L100/112/178 Please write with the Equation editor.

Response:

  Thank you for your valuable suggestion. As required, we have written all equations using the equation editor function of Microsoft Word and formatted the equations based on the target journal template for equation formats.

Comment:

L103 Please give details about the AAS method.

Response:

 We have added details regarding the water-absorption capacity in the Methods section as shown below:

The sample (2.0 g) was mixed with 50 mL of 1.0% hydrochloric acid and stirred for 1 h at 20 °C. The suspension was centrifuged at 3000 rpm for 15 min. Before measurement, the supernatant was filtered through a 0.45-μm nylon filter. Strontium chloride (0.1%) was added to the samples as an interference suppression agent for calcium and magnesium ions.

Comment:

L110 This is not the hue angle, but the color difference (delta E). Please revise in the whole manuscript and replace hue with color difference.

Response:

We apologize for the typographical error. The error has been rectified.

Comment:

L136 Please give the dilutions used.

Response:

  The sample concentrations of 156, 313, 625, 1250, 2500, and 10000 μg/mL were used for DPPH analysis; the DPPH radical scavenging activity was expressed as IC50.

Comment:

L200 Have you extracted the gluten or have you analyzed the dough? Please clarify.

Response:

 Thank you for the opportunity to provide clarity here. Gluten was not extracted in this experiment. Instead, we analyzed the gluten in dough.

Comment:

Please explain why from section 3.4 you chose only 1 treatment condition for each addition. Furthermore, in the sensory analysis, you included more samples that in the sections 3.4-3.6. Why?

Response:

  This is because the loaf volume roasting MLP at 170℃ for 20 min revealed the most promising result. Hence, this condition was used for further evaluation. This has also been mentioned in the text. Regarding sensory evaluation, we conducted preliminary experiments to identify the optimal conditions. This has also been mentioned in the Methods section.

 “Based on the results of the pilot study, the tested roasting conditions were reduced to five for the sensory test, considering the differences between the samples and the ease of comparison: untreated MLP and RMLP obtained by roasting for 20 min at 110 °C, 130 °C, 150 °C, or 170 °C.”

Comment:

Please mention the limits and further research perspectives.

Response:

  We have added the following text from “These findings underscore the importance of heat treatment in mitigating the effects of MLP on bread fermentation and swelling. In the future, it will be necessary to examine in detail the changes in components caused by heating”.

Thank you again for your comments on our paper.

Reviewer 3 Report

The manuscript is prepared thoroughly, presenting properly designed experiment. The introduction leads well to the topic and justifies accurately the undertaken research. Materials are usually detailed enough (minor remarks below). Discussion is properly lead, without any speculation or off topic areas. The conclusions are supported by the results and are clear and easy to understand. My greatest concern is related to antioxidant and polyphenol analysis. I do not find link with the research questions that need to be answered. This is proven by lack of inclusion of those results in conclusions. In other words why this part of research was included? Nevertheless, the submitted manuscript is valuable contribution to the field.

Minor remarks

Line 102-105 please provide citation of the metod or provide some details, acid used, possible matrix modifiers, oxidant (was N2O used for Ca?)

Line 110 absolute color change is common term

Line 125 please use term compression or deformation instead of strain

Author Response

Dear Reviewer 3:

We are pleased to submit a revised version of our manuscript, “Effects of roasting on the quality of Moringa oleifera leaf powder and loaf volume of Moringa-oleifera-supplemented bread” (foods-2650408).

A point-by-point response has been provided for each query, and the corresponding changes in the main manuscript have been highlighted in yellow.

Comment:

The manuscript is prepared thoroughly, presenting properly designed experiment. The introduction leads well to the topic and justifies accurately the undertaken research. Materials are usually detailed enough (minor remarks below). Discussion is properly lead, without any speculation or off topic areas. The conclusions are supported by the results and are clear and easy to understand. My greatest concern is related to antioxidant and polyphenol analysis. I do not find link with the research questions that need to be answered. This is proven by lack of inclusion of those results in conclusions. In other words why this part of research was included? Nevertheless, the submitted manuscript is valuable contribution to the field.

Response:

Thank you for the encouraging feedback. Although the purpose of this experiment was to improve the swelling of bread, we presume that the roasting process might destroy functionalities, such as antioxidant activity. We are currently analyzing the components altered on heating MLP and conducting further studies to explore this information; we intend to publish the findings as a research paper in future.

Comment:

Line 102-105 please provide citation of the metod or provide some details, acid used, possible matrix modifiers, oxidant (was N2O used for Ca?)

Response:

 Based on your suggestion, we have added the following text: The sample (2.0 g) was mixed with 50 mL of 1.0% hydrochloric acid and stirred for 1 h at 20 °C. The suspension was centrifuged at 3000 rpm for 15 min. Before measurement, the supernatant was filtered through a 0.45-μm nylon filter. Strontium chloride (0.1%) was added to the samples as an interference suppression agent for calcium and magnesium ions..

Comment:

Line 110 absolute color change is common term

Response:

  We apologize for the error. The same has been rectified.

Comment:

Line 125 please use term compression or deformation instead of strain

Response:

  We acknowledge your valuable suggestion. However, the texture analyzer we used defines this as "70% strain", hence the term.

Once again, we thank you for the thoughtful suggestions and insights, which have enriched the manuscript and produced a better and more balanced account of the research.

Round 2

Reviewer 1 Report

The manuscript has been modified according to modification suggestions.

Reviewer 2 Report

The paper can be published now,